# Genes Associated with Foliar Resistance to Septoria Nodorum Blotch of Hexaploid Wheat (*Triticum aestivum* L.)

**DOI:** 10.3390/ijms22115580

**Published:** 2021-05-25

**Authors:** Dora Li, Esther Walker, Michael Francki

**Affiliations:** 1State Agricultural Biotechnology Centre, Murdoch University, South St, Murdoch, WA 6150, Australia; D.Li@murdoch.edu.au (D.L.); esther.walker@murdoch.edu.au (E.W.); 2Department of Primary Industries and Regional Development, 3 Baron Hay Ct, South Perth, WA 6151, Australia

**Keywords:** septoria nodorum blotch (SNB), wheat, quantitative trait loci (QTL), resistance, international wheat genome sequencing Consortium (IWGSC)

## Abstract

The genetic control of host response to the fungal necrotrophic disease Septoria nodorum blotch (SNB) in bread wheat is complex, involving many minor genes. Quantitative trait loci (QTL) controlling SNB response were previously identified on chromosomes 1BS and 5BL. The aim of this study, therefore, was to align and compare the genetic map representing QTL interval on 1BS and 5BS with the reference sequence of wheat and identify resistance genes (*R*-genes) associated with SNB response. Alignment of QTL intervals identified significant genome rearrangements on 1BS between parents of the DH population EGA Blanco, Millewa and the reference sequence of Chinese Spring with subtle rearrangements on 5BL. Nevertheless, annotation of genomic intervals in the reference sequence were able to identify and map 13 and 12 *R*-genes on 1BS and 5BL, respectively. *R*-genes discriminated co-located QTL on 1BS into two distinct but linked loci. *NRC1a and TFIID* mapped in one QTL on 1BS whereas *RGA* and *Snn1* mapped in the linked locus and all were associated with SNB resistance but in one environment only. Similarly, *Tsn1* and *WK35* were mapped in one QTL on 5BL with *NETWORKED 1A* and *RGA* genes mapped in the linked QTL interval. This study provided new insights on possible biochemical, cellular and molecular mechanisms responding to SNB infection in different environments and also addressed limitations of using the reference sequence to identify the full complement of functional *R*-genes in modern varieties.

## 1. Introduction

Septoria nodorum blotch (SNB) is a fungal disease of wheat caused by the necrotrophic pathogen *Parastagonospora* (syn. ana, *Stagonospora*; teleo, *Phaeosphaeria*) *nodorum* (Berk.). Symptoms of infection on the leaf include lens-shaped red-brown lesions surrounded by a yellow halo that coalesce to form blotches with pycnidia evident in time. Disease symptoms on glumes include dark brown to purple with ash-grey areas embedded with black pycnidia, and become apparent following distinct symptoms on leaves. SNB can cause significant yield loss of up to 12.9% in Western Australia (WA) [1] due to the reduced photosynthetic capacity caused by infection in the leaves and glume [2,3,4].

Reducing SNB severity in adult plants is important when the disease is at its greatest in-season risk during warmer spring conditions in WA. Therefore, a number of studies have focused on evaluating SNB response of bi-parental homozygous populations and global germplasm collections in WA environments to unravel the genetic basis of adult plant resistance [5,6,7]. Analysis of bi-parental mapping populations detected similar quantitative trait loci (QTL) in successive years in WA on chromosomes 1B and 2A [6] and 2D [5,6,7], whereas environment-specific QTL were detected on 4B, 5A and 5B [5,6]. Significant genotype-by-environment-by-isolate interactions causing variable disease response was further supported when a collection of 232 global wheat accessions were evaluated in multiple environments in WA using genome-wide association study (GWAS) [7]. A comparison of genetic response to SNB infection in Nordic wheat confirmed genotype-by-environment interactions and environment-specific QTL in other regions of the world under natural disease infection [8,9,10]. Alternative host genes, infection by genetically diverse isolates with variability in virulence or aggressiveness, or a combination of factors may cause different SNB responses of wheat genotypes when evaluated in different environments [7].

Knowledge on biological mechanisms involved with SNB response has been limited until recently when studies identified genetic and biochemical interactions between host and pathogen. *Parastagonospora nodorum* produces a range of necrotrophic effector (NE) proteins with an inverse gene-for-gene relationship with the host contributing towards plant necrosis [11]. There are a range of NEs discovered with corresponding sensitivity loci mapped in wheat including SnToxA–*Tsn1* interaction on chromosome 5B [12,13], SnTox1–*Snn1* on 1B [14,15], SnTox2–*Snn2* on 2D [13], SnTox3–*Snn3-B1* on 5B [16,17], SnTox4–*Snn4* on 1A [18], SnTox5–*Snn5* on 4B [19], SnTox6–*Snn6* on 6A [20] and SnTox7–*Snn7* on 2D [21]. The relationship between NE and disease response in multiple environments was realized through a comparison of chromosomal map locations for NE-*Snn* loci and QTL. The association of *Tsn1* locus, for example, was environment-specific [6,7,22], indicating that specific SnTox–*Snn* interactions may have limited influence in controlling disease progression in some environments. Similar inferences were made in GWAS analysis when known *Snn* loci were in linkage equilibrium with QTL for foliar and glume response in several environments [7,10]. The limited influence of known NE–*Snn* interactions regulating SNB response in different environments was perpetuated from field evaluation in Norway, the United Kingdom and the eastern region of the US [10,23]. Therefore, it appears that host genes other than known *Snn* and *Tsn1* loci contribute towards disease response against different isolates and environments.

Despite known NE–*Snn* interactions being disparate with SNB disease development across multiple environments and diverse isolates [7,9,10,23], the cloning of *Tsn1* and *Snn1* genes provided initial insights into biological mechanisms underpinning response to NE and/or SNB disease. The protein translated by *Tsn1* has typical disease resistance features including serine threonine protein kinase and nucleotide binding site-C-terminal leucine rich repeat (NBS-LRR) domains [24], whereas *Snn1* encodes a wall-associated kinase (*WK*) [15]. These genes and the corresponding proteins are presumed to exploit plant defence mechanisms resembling effector-triggered immunity (ETI) and pathogen-associated molecular pattern (PAMP)-triggered immunity (PTI) [25]. The proteins are thought to hijack signalling pathways for resistance to biotrophic pathogens rendering susceptibility of the host to SNB infection [15,26]. It is feasible, however, that *P. nodorum* infection can also exploit other aspects of ETI/PTI pathways or, indeed, alternative host defence mechanisms. Various biochemical and cellular components provide alternative biological targets for necrotrophs to penetrate host defences and elicit a disease response during pathogen invasion [27].

Whilst chromosome walking and physical assembly was used to clone *Tsn1* and *Snn1*, the release of a high-quality reference sequence for wheat [28,29] may expedite discovery of other genes responding to SNB. Coverage of up to 94% of a reference genome sequence of hexaploid wheat from the experimental line Chinese Spring and subsequent annotation identified 104,091–107,891 protein coding genes [29,30] but did not necessarily represent an absolute value of gene copy number across bread wheat cultivars. Copy number variants (CNV), presence–absence variants (PAV) and single nucleotide polymorphisms (SNP) contributed towards genomic differences amongst modern wheat cultivars with an average of >128,500 genes per cultivar estimated from pan-genome sequencing [31,32]. Despite structural variation in genomes across wheat cultivars, the public availability of a reference genome provides an initial basis to discover genes and alleles that may underpin trait variation.

The aim of this study, therefore, was to align and compare the genetic map representing QTL interval for SNB foliar resistance on chromosome 1BS and 5BL [6,33] with the physical map and identify resistance genes (*R*-genes) associated with SNB response. Doing so will identify similarities or structural differences between QTL intervals for EGA Blanco, Millewa and the reference sequence, with potential for discovery of alternative genes and proteins of PTI/ETI other than the known genes *Snn1* and *Tsn1* or other biological pathways. Identification of genes encoding proteins with similarity to disease resistance genes (*R*-genes) having significant association with SNB response would provide wider insight into biological pathways involved in SNB response and their interactions with different isolates and environmental cues.

## 2. Results

### 2.1. Alignment of Genetic and Physical Maps and Gene Annotation on Chromosomes 1BS and 5BL

The genetic maps with QTL on chromosomes 1BS and 5BL [33] were aligned to the IWGSC reference sequence (Gramene release #62, November 2019) using DNA sequence identity of mapped SNP markers. The genetic order of markers within and flanking the QTL interval on chromosome 1BS did not align with the physical order of genes with small intrachromosomal translocations apparent between the genomes of EGA Blanco, Millewa and the reference genome sequence (Figure 1A).

Analysis of the physical region on 1BS, therefore, was extended beyond the QTL delineated by markers *wPt-8267* (SNP at 1,825,338 bp) and *IWB64368* (SNP at 4,320,856 bp) to include markers *IWB7273* (SNP at 1,203,968 bp) and *IWB26464* (SNP at 22,701,051 bp) in the reference genome (Figure 1A) to account for potential *R*-gene content flanking the interval in the reference genome but may be represented within the QTL in the genetic map. Annotation of the 21.5 Mbp region on 1BS identified a total of 423 genes with 122 having significant identity with proteins encoded by *R*-genes in the reference sequence (Appendix A). Interestingly, multiple copies of *R*-gene families were apparent in the 21.5 Mbp region, which included *NRC* (3 copies), *RPM1* (25 copies), *Lr21* (3 copies), *Pm3* (9 copies), *LRR* (26 copies) and *RPP13* (27 copies) (Appendix A). Furthermore, the gene *TraesCS1B02G004100.1* at 2,354,630 bp had 100% amino acid identity with *Snn1* gene known to be involved with SNB response (Appendix A). It is evident, therefore, the targeted region on chromosome 1BS contained highly duplicated members of *R*-gene families interspersed amongst other low copy defence response and non-disease related sequences in the reference sequence of Chinese Spring with apparent rearrangements compared with the genomes of EGA Blanco and Millewa. Annotation of disease and non-disease related genes, their physical location including and encoded amino acid residues and identity with annotated proteins is provided in Appendix A.

The genetic map of chromosome 5BL that harboured two environment-specific QTL for SNB [6,33] was aligned to the physical map using flanking SNP markers *IWB12583* (SNP at 539,072,424 bp) and *IWB27379* (SNP at 580,686,141 bp) (Figure 1B). Markers with a physical location in the reference genome sequence were congruent with genetic map position indicating no substantial gene or genome rearrangements for this region in EGA Blanco or Millewa (Figure 1B). Moreover, the genetic distance of 41.7 cM between *IWB12583* and *IWB27379* encompassing the two QTL for SNB resistance represented a physical distance of 45.6 Mbp (Figure 1B) containing 448 genes with their physical location on the Chinese Spring reference genome provided in Appendix A. A total of 39 genes were annotated as having significant identity with proteins encoded by *R*-genes or wall-associated kinase genes known to be involved with disease response. Some multi-gene families were identified on chromosome 5BL, including two copies having identity with known disease resistance genes *RPP13*, *RPM1*, *Lr10* and peroxidase (*Per*), four copies of *LRR* and 11 copies of wall-associated kinases (*WAK*) (Appendix A). The remaining 16 *R*-genes were r single copy within the 45.6 Mbp region (Appendix A). Interestingly, annotation of the 41.7 cM physical region on 5BL did not identify the location of the *Tsn1* gene involved in NE interactions [24], confirming that this gene was missing within the QTL interval of the Chinese Spring reference genome.

### 2.2. Genes on Chromosome 1BS Associated with Foliar SNB Resistance

Members of *R*-gene families (Appendix A) were specifically selected at regular intervals spanning the 21.5 Mbp region, alleles identified either through PAV or from gene re-sequencing and subsequent markers genetically mapped to refine QTL regions associated with foliar SNB response to a smaller physical region. The smaller intervals were subsequently used to target further *R*-genes within the QTL. PCR primers for tiling paths were designed from a total of 62 *R*-genes extracted from the reference genome representing members of *RPP13*, *RPM1*, *NRC*, *Pm3* and *Lr21*, *RGA*, *LRR* and *RR* gene families (Appendix A). Collectively, 27 genes were polymorphic and markers were developed based on PAV and mapped in the EGA Blanco/Millewa population (Table 1; Figure 2A).

The remaining 35 selected genes had neither amplicon size differences nor DNA sequence polymorphism between parental varieties, and therefore were presumed not to be associated with SNB response. Analogous to the iSelect Infinium 90K markers used to align genetic and physical maps (Figure 1A), genetic order for 27 *R*-gene loci did not align with physical order (Table 1; Appendix A). Assuming the physical sequence assembled in Gramene Release #62 is accurate, it appears that *R*-gene translocations were apparent on chromosome 1BS between genomes of EGA Blanco and Millewa compared with the reference sequence from Chinese Spring.

The co-located QTL on chromosome 1BS, *QSnl07.daw-1B* and *QSnl08.daw-1B*, assumed that the same gene controlled SNB foliar resistance in different environments [6,33]. The genetic linkage map consisting of SNP markers from the iSelect 90K genotyping array [33] were integrated with markers developed from *R*-genes in this study and subsequent QTL analysis confirmed two distinct and environment-specific QTL from field phenotyping data in 2007 and 2008 (Figure 2A). Therefore, 1BS contained several genes at two loci that responded independently to SNB. Genes *TraesCS1B02G002400.1* with identity to *NBS-LRR REQUIRED FOR HYPERSENSITIVE RESPONSE-ASSOCIATED CELL DEATH 1* (*NRC 1a*; AKC34064.1) (Appendix A) represented by the marker *1BR_NRC_5A* (Table 2) and *TraesCS1B02G007800.1* having identity with a The *Transcription Initiation Factor sub-unit II* (*TFIID*; EMT21817.1) (Appendix A) represented by the SNP marker *IWB64368* (Table 2) had highly significant (*p* < 0.001) association with *QSnl07.daw-1B* by way of LOD scores of 13.6 and 18.6, respectively (Table 2; Figure 2A).

Neither of the genes, however, were associated (*p* > 0.05) with *QSnl08.daw-1B* (Figure 2A), indicating that their contribution to SNB response was specific to the 2007 environment. Instead, gene *TraesCS1B02G003100.1* with identity to a putative *Resistance Gene Analogue* (*RGA*; KAE8799430.1) represented by *1BR_RGA_4B*, and *Snn1* (*TraesCS1B02G004100.1*) represented by *X3476283* (2.35 Mbp) (Appendix A), had highly significant association (*p* < 0.001) with *QSnl08.daw-1B*, with LOD scores of 10.0 and 10.1, respectively (Table 2; Figure 2A). Therefore, two genes up to 2.54 Mbp apart on chromosome 1BS had significant association with SNB resistance in each environment.

### 2.3. Genes on Chromosome 5BL Associated with Foliar SNB Resistance

Similar to the analysis of genes on chromosome 1BS, *R*-genes were selected at regular intervals spanning the 45.6 Mbp region on chromosome 5BL, mapped and further genes identified in Gramene Release#62 from within refined QTL region. A total of 39 genes in the 45.6 Mbp region encoding proteins with highest similarity to disease resistance and wall-associated kinases were analysed for allele differences, either by amplicon size differences or SNP discovery (Appendix A). There were six annotated *R*-genes with either amplicon size difference or SNP between parental varieties EGA Blanco and Millewa (Table 3).

The remaining genes showed no DNA sequence polymorphism and were therefore not further considered to be associated with SNB resistance. The markers *Xfcp623 and Xfcp620*, representing *Tsn1* and cell wall-associated kinase (*WK35*) genes, respectively [24], were also mapped at 234.9 cM (Table 3) despite the former gene not being represented in the physical interval of 45.6 Mbp on 5B of the wheat genome reference sequence. The 14 genes with PAV or SNP were assigned to the EGA Blanco/Millewa genetic map and the genetic order between 234.9 cM and 270.2 cM (Figure 2B) aligned with the physical map between 546.56 Mbp and 589.16 Mbp of the reference genome sequence from Chinese Spring (Table 3).

QTL mapping using the genetic map with ordered *R*-genes showed two distinct regions for SNB resistance on chromosome 5BL detected when the DH population was evaluated for SNB foliar response in 2007 and 2008. The position of QTL (Figure 2B) were similar to that previously reported [6,33]. One-dimensional scanning positions of genes associated with QTL identified *Xfcp620 (WK35)* and *Xfcp623* (*Tsn1*) at the same genetic locus (Table 2; Figure 2B), having a significant association (*p* < 0.001) with SNB resistance with a LOD score of 7.5 when the population was evaluated in 2007 (Table 2). However, neither *WK35* nor *Tsn1* were associated with SNB response in 2008 (Table 2). Instead, the SNP marker *IWA4103* at 558.62 Mbp representing gene *TraesCS5B02G380700.1* encoding a *NETWORKED 1A* protein (XP_003577773.1) (Appendix A) and *5B_RGA_SNP_3A1* at 562.79 Mbp representing gene *TraesCS5B02G383800.1* encoding a putative *RGA* (XP_020196799.1) (Appendix A) had significant (*p* < 0.001) associations with SNB resistance (LOD scores of 2.8 and 4.3, respectively) when evaluated in 2008 (Table 2). Therefore, it appears that at least three genes in addition to *Tsn1* on chromosome 5BL are associated with SNB infection, but in different environments.

### 2.4. Gene Structure Analysis

A structural comparison of genes associated with SNB resistance on chromosomes 1BS and 5BL was carried out to reveal differences that could explain variable SNB response between EGA Blanco and Millewa. The annotation of four genes retrieved from the genome assembly of chromosome 1BS and associated with SNB response in one of two environments and subsequent design of tiling paths provided opportunities to re-sequence and identify alleles from cultivars EGA Blanco and Millewa (Figure 3).

No tiling paths from *NRC1a*, *RGA* and *Snn1* amplified PCR products from the SNB susceptible cultivar, Millewa, whereas most tiling paths for the *TFIID* gene were able to be amplified from the resistant cultivar, EGA Blanco (Figure 3). It is assumed, therefore, that gene loss in the QTL for SNB foliar response on chromosome 1BS was apparent in the genome of cultivars EGA Blanco and Millewa compared to Chinese Spring. Amplification of tiling paths from *Snn1* of cultivar Millewa showed identical exon and intron size with the corresponding gene in the reference genome. The remaining genes, however, either showed near complete intron–exon structure (*TFIID*) or partial gene sequence (*NRC1a* and *RGA*) (Figure 3) due to the inability to amplify specific tiling paths based on primers designed against corresponding genes from the Chinese Spring reference genome. Therefore, deletion, or significant gene sequence variation, is apparent on EGA Blanco and Millewa residing within the QTL interval on chromosome 1BS.

The four candidate *R*-genes having significant association with foliar SNB response on chromosome 5BL were amplified and characterized from EGA Blanco and Millewa. All tiling paths from *NETWORK1A* and *RGA* genes were amplified from both EGA Blanco and Millewa with similar exon sizes to those extracted from the reference genome (Figure 4).

The sequence difference between EGA Blanco and Millewa for *NETWORK1A* was subtle, with only one SNP identified in exon 6 (Figure 4A) whereas *RGA* identified six SNPs and an INDEL of 1 bp in intron 1 between EGA Blanco and Millewa (Figure 4B). The inability to amplify tiling paths for *Tsn1* confirmed the gene was deleted in EGA Blanco (Figure 4C) whereas only partial sequence was available for *WK35* indicating considerable gene variation between EGA Blanco and Millewa (Figure 4D).

## 3. Discussion

The availability of high-density SNP array and high-quality reference genome sequence provided a foundation to compare genetic and physical maps for the QTL intervals on chromosomes 1BS and 5BL as a means to identify underlying candidate *R*-genes associated with SNB response. Neither EGA Blanco nor Millewa contained known alien translocated segments in their genomes determined by PCR analysis for species-specific repetitive sequences (data not shown), therefore, small scale translocations between genomes of EGA Blanco and Millewa with respect to the reference genome are likely to reflect intrachromosomal rearrangements during domestication of these Australian cultivars. Cytogenetic analysis [34,35] and high-density SNP arrays identified significant large and small translocations, deletions and inversions in different cultivars and breeding lines when compared with the reference genome sequence [32,35], and postulated that chromosome rearrangements and complex breeding histories provided lineage divergence for selective advantages in diverse environments. In contrast, the alignment of markers between genetic and physical maps for the region on 5BL did not detect large-scale chromosomal rearrangements between genomes of EGA Blanco, Millewa and the Chinese Spring reference sequence, indicating that some regions of genomes may harbour smaller structural variants or more conserved regions required for selective adaptation of modern varieties to broader environments. It is now becoming increasingly evident that structural variants, whether large or small-scale (including CNV and PAV), between genomes of breeding lines are associated with responses to biotic and biotic stresses [36].

Small structural variants such as CNV and PAV were common between the genome of the Chinese Spring reference sequence and cultivars EGA Blanco and Millewa. Multiple copies of *R*-gene families on chromosome 1BS and 5BL indicated that gene duplication was evident during the evolution of Chinses Spring reference genome, with opportunities to explore representation of specific gene family members and variation between the two Australian cultivars within QTL regions for SNB response. Despite the presence or absence of markers to specific regions for 13 and 12 *R*-gene members around QTL on 1BS and 5BL, respectively, it was inconclusive whether the nature of structural variants between the reference sequence and the genomes of both cultivars represented entire gene loss or gain, insertion/deletions (INDEL) of specific gene regions or significant sequence diversity at PCR primer sites that precluded successful amplification of tiling paths. Nevertheless, it appeared there is significant structural variation for genes within and flanking QTL between EGA Blanco and Millewa. Recent evidence for high rates of homologous gene duplication and deletion events for members of the Bowman–Birk inhibitor gene family across varieties examined from pan-genomes [37] is testament to the dynamic nature of the wheat genome, leading to gene diversification and adaptability against biotic and abiotic stress during domestication and breeding.

This study further explored the diversification of specific gene copies between the reference sequence and genomes of EGA Blanco and Millewa for eight genes associated with SNB response on 1BS and 5BL. The lack of amplification of all tiling paths for five of the eight genes in either EGA Blanco or Millewa made it reasonable to assume that whole genes representing *NRC1a (**TraesCS1B02G002400.1)*, *RGA (**TraesCS1B02G003100.1)*, *Snn1* (*TraesCS1B02G004100.1*) and *TFIID* (*TraesCS1B02G007800.1*) on 1BS and *Tsn1* (GU259618.1) on 5BL were deleted in some varieties during domestication and breeding. Genes that were retained in only one cultivar (except *Snn1*) or in both, such as *WK35* (*TraesCS5B02G368200.1*), only had regions with high identity to corresponding genes on 1BS and 5BL in the reference sequence, whereas *NETWORKED 1A* (*TraesCS5B02G380700.1*) and *RGA A* (*TraesCS5B02G383800.1*) were retained in both cultivars with subtle rearrangements. Therefore, it appeared that continual evolution during domestication from a common progenitor line resulted in an array of gene rearrangements across the two cultivars and the reference sequence. Rearrangements included an assortment of PAV, INDEL and SNP. There is increasing evidence that intrachromosomal variation between wheat varieties are common and gene sequences in breeding lines may not be represented in Chinese Spring [31,32,35,38], highlighting the limitations of using a single reference sequence to obtain a complete functional gene complement across breeding lines and cultivars. Therefore, it is possible that additional gene candidates within the QTL on 1BS and 5BL other than those identified in this study were represented in EGA Blanco and Millewa but not in Chinese Spring. In that regard, presence of the *Tsn1* gene in Millewa but absent from the corresponding region in Chinese spring exemplify limitations when using a single reference sequence to identify genes controlling trait variation within QTL intervals in breeding lines and cultivars. Therefore, research efforts are moving from relying on a single reference genome in crop species to resequencing multiple accessions in an attempt to capture a wider repertoire of gene diversity as a pan-genome [36,39]. A complete genome sequence for EGA Blanco and Millewa will add to the wheat pan-genome [31,32] and further contribute towards identifying the entire functional gene complement controlling SNB response on 1BS and 5BL.

High-density SNP markers were unable to discriminate QTL controlling foliar SNB on 1BS, so it was presumed that similar genes responded to SNB foliar infection in at least two environments [33]. However, increasing genetic mapping density within the QTL interval using *R*-gene markers in the EGA Blanco and Millewa DH population determined that two distinct QTL reside on 1BS where SNB response appear to be controlled by different genes ~7 cM apart. Environment-specific QTL is consistent with recent reports for foliar SNB response on 1B and other wheat chromosomes based on disease evaluation in different WA locations [7] and other regions of the world [10,40]. Interestingly, *Snn1* [14,15] was only associated with QTL in 2008 (*QSnl08.daw-1B*) but not in 2007, indicating variability in SnTox1–*Snn1* interactions in WA environments, similar to that reported for SnToxA–*Tsn1* interactions on 5BL [6,7]. We expected that marker intervals for linked QTL on 5BL would be reduced using gene markers, however, the number of *R*-genes relative to the total gene content in the annotated regions of the reference sequence was significantly lower on 5BL than 1BS (8.7% and 27.2%, respectively). All 39 *R*-genes from a total of 448 genes on 5BL were investigated for marker development but none were identified as polymorphic (PAV, INDEL or SNP) for association analysis within the *QSnl07.daw-5B* region. Analysis of the entire functional gene complement by whole genome sequence of EGA Blanco and Millewa would provide further information on *R*-genes not represented in the reference sequence, if any, that may be associated with *QSnl07.daw-5B* and, indeed, *QSnl08.daw-5B.*

Quantitative plant disease resistance is a complex multicomponent system [41] so it is expected that a range of genes associated with SNB response are likely to involve components of biochemical, cellular and molecular pathways. Despite the unlikelihood of an entire functional gene complement being identified on 1BS and 5BL using a single reference sequence, a number of genes other than *Tsn1, WK35* and *Snn1* [15,24] were identified which allowed further insights into alternative biological mechanisms. Disease resistance gene analogues (*RGA*) represent broad gene families including NBS-LRR and transmembrane-LRR, both of which contain a number of classes and sub-classes ubiquitous in plant genomes that function directly in binding effector proteins or modifying host proteins to activate a cascade of signal transduction pathways [42]. The association of diverged copies of *RGA* genes on 1BS and 5BL indicated functional variation related to pathogen recognition or other signalling pathways in response to SNB in different environments, but an accurate prediction of their role in that regard is yet to be determined. Whole genome sequence of EGA Blanco and Millewa will provide further evolutionary clues leading towards unravelling functional significance of those variants. Likewise, genes related to signalling pathways were extended to include *NRC1* that encode critical protein nodes for pattern and effector mediated signalling for cell death in response to pathogen infection [43,44,45,46]. The significance of either diverged or deleted RGA and *NRC1* genes for mediating signalling pathways in environment-specific SNB response is not yet apparent but warrants further experimental investigation.

Signal transduction pathways also include those involved at the intracellular–extracellular boundary during fungal infection [27,47], with evidence that genes associated with SNB may be involved through basal defence mechanisms. *NETWORKED 1A* is a multigene family encoding actin-binding proteins to form endoplasmic reticulum and plasma membrane complexes that react to extracellular signals, including biotic stresses [48,49]. Subtle variation of a member of the *NETWORKED 1A* gene between resistant (EGA Blanco) and susceptible (Millewa) varieties may lead to protein differences that mediate actin–membrane interactions at the plasma membrane–cell wall interface after fungal infection influencing pathogen spread and onset of disease. The specific role of proteins encoded by *NETWORKED 1A* alleles would shed further light on possible function for components of basal defence mechanisms in reducing or enhancing SNB disease progression.

Plant response to pathogen infection relies on complex mechanisms of gene regulation for components of pathogen recognition, signal transduction and defence. The *TFIID* gene associated with SNB response on 1BS indicated molecular elements may regulate pathogen infection and disease response through specific gene expression in biochemical and cellular pathways. *TFIID* recognise and bind to core promoter elements (generally TATA boxes) in a complex assembly of transcription machinery that interact with co-factors, gene-specific activators and repressors for certain levels of transcriptional output [50,51]. Knowledge on specific roles for *TFIID* in plant disease response is limited but differential gene expression in response to crown rot in wheat [52] and association with Cercospora leaf spot disease in QTL mapping in mungbean [53] indicated that variation in *TFIID* may regulate transcriptional activity for specific genes of disease response pathways. However, further knowledge is required on the specific genes regulated by *TFIID* and how sequence variation contributes to pathogen recognition, signal transduction and defence towards eliciting an SNB response.

Although structural variation exists between hexaploid wheat genomes of different origins, exploitation of the wheat reference sequence in this study provided a preliminary basis to target some, but unlikely all, genes controlling SNB response. Identifying the full complement of functional genes would require resequencing genomes of EGA Blanco, Millewa and indeed, other modern varieties and breeding lines contributing towards a pangenome that will unravel the complex functional and coordinated roles of biochemical, cellular and molecular mechanisms that elicit a disease response when wheat is challenged with diverse SNB isolates in a specific environment.

## 4. Materials and Methods

### 4.1. Plant Material and Evaluation for SNB Response

The DH population consisting of 241 lines developed from EGA Blanco/Millewa and phenotypic evaluation in two environments at South Perth, Western Australia, in 2007 and 2008, as previously described [6], was accessed from the grains collection at the Department of Primary Industries and Regional Development. Details of mixed isolates used at both locations, trial designs and methodology for inoculation and evaluating disease response, phenotypic and statistical analysis of parents and individuals from the EGA Blanco/Millewa DH population and QTL detection using mean plot values for each individual were previously described [6]. Mean plot SNB scores from 2007 and 2008 disease evaluation [6] were used for QTL analysis in this study to identify genes associated with SNB response. Genotyping data and map construction with high-density SNP markers was previously described [33].

### 4.2. Aligning Genetic and Physical Maps and Annotation of Gene Content

The genetic map and QTL intervals for SNB resistance on chromosomes 1BS and 5BL [33] were aligned with the whole genome assembly of bead wheat (Refseq v1.0) through the IWGSC database (Gramene release #62 November 2019 at http://www.gramene.org accessed on 24 April 2021). DNA sequence of SNP markers within and flanking each QTL on chromosomes 1B and 5B (Figure 1) were used as query in BLASTN analysis. Hits having >99% identity with sequences in the reference genome demarcated the physical interval for each QTL. Prediction of coding and non-coding genes within the physical interval were accessed from the de novo wheat genome assembly and further annotated using BLASTP and SmartBLAST alignment against database repository from NCBI (https://blast.ncbi.nlm.nih.gov/Blast.cgi accessed on 24 April 2021). Protein similarity was determined with an amino acid identity threshold of >50% identity for >50% sequence coverage (*e* value < 10^−5^). Genes encoding proteins having amino acid identity to known disease resistance genes, disease resistance gene motifs or defence response pathways were considered as candidate *R*-genes (Appendix A).

### 4.3. Primer Design and Tiling Paths

Sequences from each of the candidate genes within physical intervals from either chromosome 1BS or 5BL were extracted and subjected to BLASTN of the whole genome assembly (Gramene release #62 November 2019 at http://www.gramene.org accessed on 24 April 2021) to identify homologs on the A, B and D genomes. DNA sequence alignment of gene homologs identified sub-genome specific sequence variants and PCR primers were designed to target genome and allele specific polymorphisms tiled across the total available sequence from the reference genome cv Chinese Spring IWSGC Refseq v1.0. PCR primers were designed for amplicons using Primer3Plus [54] to have a minimum tiling overlap of 100 bp. Primer sequences for tiling paths of selected genes are summarised in Appendix A.

### 4.4. PCR Amplification and Sequencing of Genomic Tiling Paths

PCR amplification was carried out in a 10 µl reaction volume containing 50 ng genomic DNA, 1U Taq DNA polymerase (Meridian Bioscience, Cincinnati, OH, USA), 0.15 mM dNTP (Meridian Bioscience, Cincinnati, USA), 1 µM primers (Macrogen Inc. Seoul, South Korea). Amplification was performed on a thermocycler (Applied Biosystems, Foster City, CA, USA) under the following conditions: 1 cycle 95 °C for 3 min, 5 cycles of 95 °C for 30 s, 65–55 °C (primer dependent) 30 s, 72 °C for 1 min, decreasing 1 °C per cycle followed by 30 cycles of 95 °C for 1 min, 60–50 °C (primer dependent) 30 s, 72 °C for 1 min with a final extension at 72 °C for 5 min. PCR products were separated on a 2% agarose gel via electrophoresis and visualised under UV light using 1X GelGreen (Biotium, Fremont, CA, USA). PCR amplifications were performed using genomic DNA extracted from the experimental line Chinese Spring and cultivars ‘EGA Blanco’ and ‘Millewa’. Genome and chromosome specificity for each candidate gene was confirmed using at least one amplicon from a tiling path when DNA from Chinese Spring and nulli-tetrasomic lines for chromosomes 1A, 1B, 1D, 5A, 5B and 5D [55] were used as a template in the PCR reaction.

Amplicons of tiling paths from cultivars EGA Blanco and Millewa were either directly sequenced or cloned into the pGEM-T easy vector system (Promega, Madison, WI, USA) following manufacturer’s instructions and sequenced using capillary electrophoresis sequencing (Macrogen Inc.). A minimum of three independent PCR reactions or cloned products was sequenced for each PCR product.

### 4.5. Gene Marker Development, Genotyping and QTL Analysis

PCR amplicons from tiling paths which resulted in a PAV between EGA Blanco and Millewa were directly mapped as candidate gene markers. The remaining tiling paths conserved between EGA Blanco and Millewa were then sequenced to identify SNP for the use in marker development.

Individuals from the EGA Blanco/Millewa DH population were genotyped for candidate gene markers (PAV or SNP-based marker) and integrated with the genetic map totalling 13,308 markers merged from the 90K iSelect SNP array, DArT and SSR marker sets, as previously described [33,56]. Candidate gene markers with less than 80% call rate or segregation distortion (*p*-value > 0.10 calculated from a χ2 test), samples with a high crossover rate (indicating a mixed sample) or duplicate samples (>95% identical genotypes) were removed from the analysis. Individuals of the DH population were also genotyped for diagnostic markers *Xfcp623*, *Xfcp620* and *X3476283* for genes *Tsn1*, *WK35* [24] and *Snn1* [15], respectively. Genetic map construction was realized using QTL IciMapping Version 4.0.6.0 [57] to determine marker order.

QTL IciMapping was used for inclusive composite interval mapping with additive effect (ICIM-ADD) using mean foliar infection scores collected at South Perth, Western Australia in 2007 and 2008, as previously described [6]. Redundant markers were removed by selecting the BIN parameter within QTL IciMapping using “Delete redundancy By Missing Rate (%)” and missing data were set at 20% using the “Delete markers By missing rate (%)” function. MAP parameters were set at LOD6, specifying Algorithm: nnTwoOpt and Rippling:SARF. Parameters for all environments were set to 0.1 cM steps, Probability in Stepwise Regression (PIN) of 0.001, minimum LOD threshold of 2.5, and deleting missing phenotypes. Final maps and associated QTL were drawn using MapChart Version 2.3 [58] and QTL IciMapping.

### 4.6. Gene Structure Analysis

DNA sequence of tiling paths amplified from wheat cultivars EGA Blanco and Millewa were assembled using Geneious R10 (https://www.geneious.com accessed on 24 April 2021, San Diego, CA, USA) and aligned with corresponding gene sequence annotations retrieved from the whole genome assembly (Refseq v1.0) with >95% identity confirmed from BLASTN analysis.

## Figures and Tables

**Figure 1 ijms-22-05580-f001:**
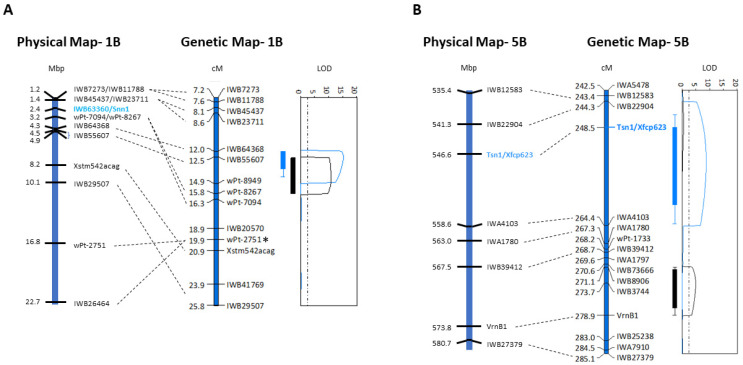
Alignment of the physical and genetic maps for QTL on chromosomes (**A**) 1BS and (**B**) 5BL. QTL traces on 1BS with corresponding LOD scores for foliar disease scores in 2007 (blue bar and lines) and in 2008 (black bar and lines) as previously reported [6,33] are shown to the right of each figure. Alignment of SNP markers within and flanking each QTL between the genetic map (in centimorgan distances, cM) and the reference sequence (in megabase pair distances, Mbp) are shown with dashed black lines. The asterisk denotes DArT marker co-segregating with SNP marker IWB26464. Markers for *Snn1* and *Tsn1* are indicated in blue.

**Figure 2 ijms-22-05580-f002:**
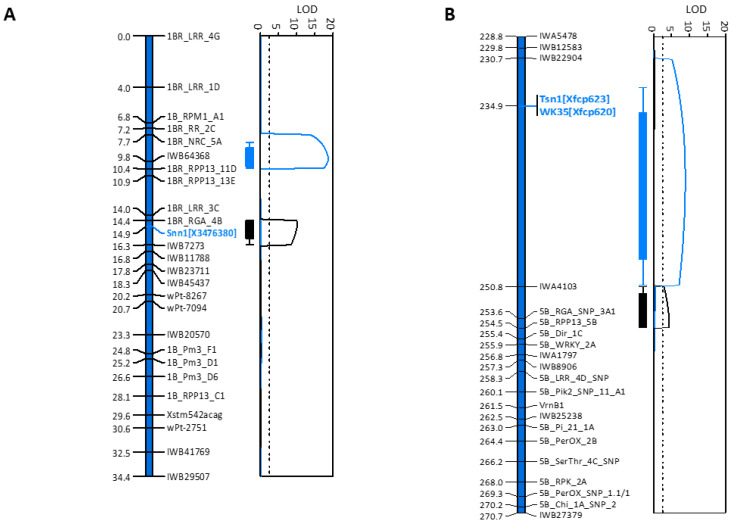
Genetic map of *R*-gene markers developed in this study integrated with genetic map of SNP and other markers [33] for chromosomes (**A**) 1BS and (**B**) 5BL. Markers for *Snn1*, *Tsn1* and *WK35* genes are highlighted in blue. QTL traces with corresponding LOD scores for foliar disease scores in 2007 (blue bar and lines) and in 2008 (black bar and lines) on 1BS and 5BL as previously reported [33] are shown to the right of each figure.

**Figure 3 ijms-22-05580-f003:**
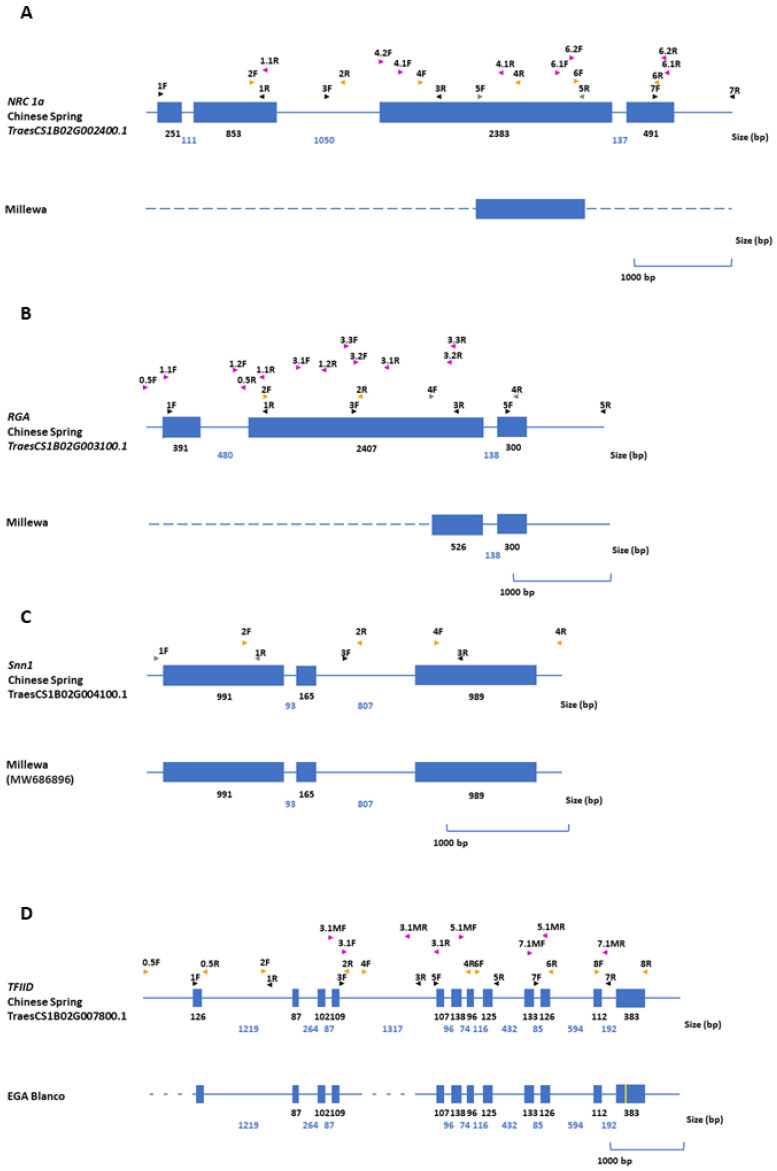
Gene structure for (**A**) *NRC1a*, (**B**) *RGA*, (**C**) *Snn1* and (**D**) *TFIID* on chromosome 1BS from reference sequence and varieties. Blue lines and boxes represent introns and exons, respectively, for annotated genes from the reference sequence. Position of PCR primers for amplifying tiling paths from varieties are shown above gene structure from the Chinese Spring reference sequence. Size of introns and exons are shown in base pairs (bp). Dashed blue lines represent regions where tiling paths were unable to be amplified in corresponding varieties. Yellow lines represent SNP between Chinese Spring and varieties. Genbank accession numbers for complete annotated genes are provided in parentheses.

**Figure 4 ijms-22-05580-f004:**
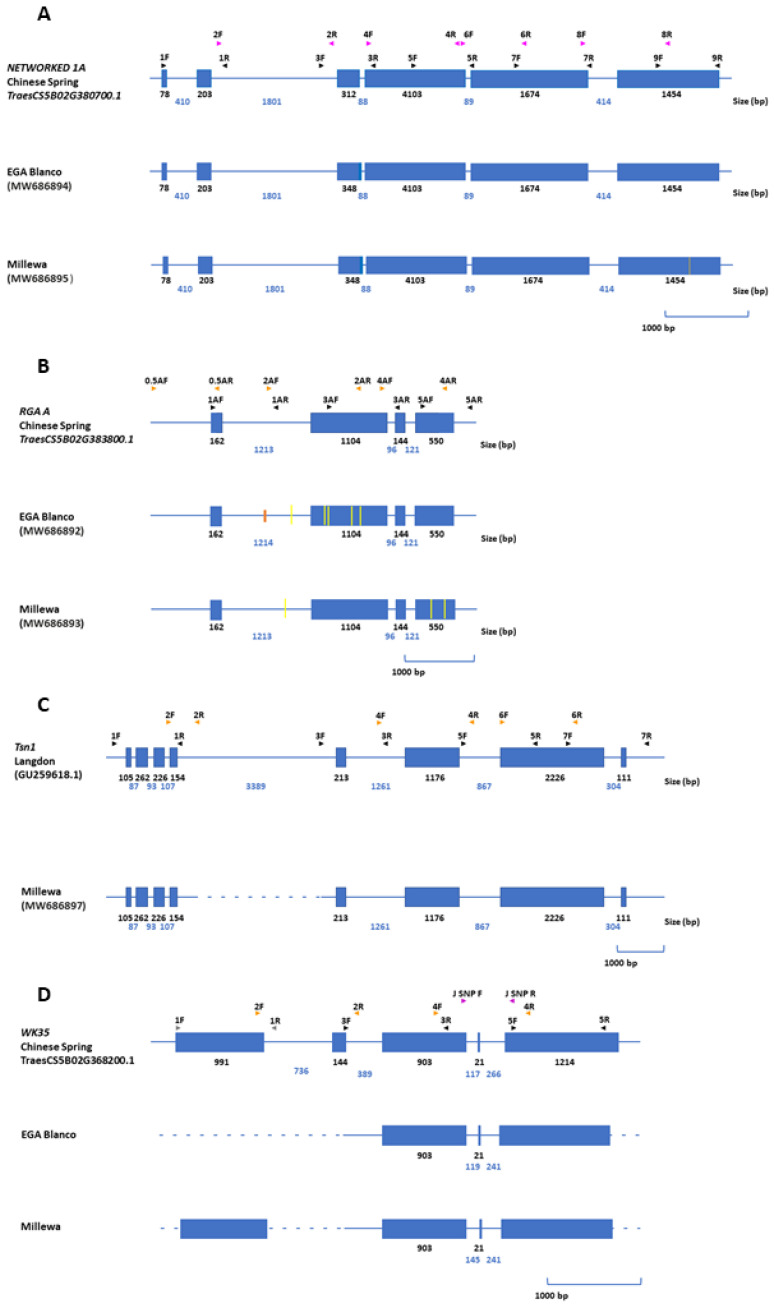
Gene structure for (**A**) *NETWORKED 1A*, (**B**) *RGA* A, (**C**) *Tsn1* and (**D**) *Wk35* on chromosome 5BL from reference sequence and varieties. Blue lines and boxes represent introns and exons, respectively, for annotated genes from reference sequences. Position of PCR primers for amplifying tiling paths from varieties are shown above gene structure from the reference sequences. Size of introns and exons are shown in base pairs (bp). Dashed blue lines represent regions where tiling paths were unable to be amplified in corresponding varieties. Yellow and orange lines represent SNP and INDELS, respectively, between reference sequences and varieties. Genbank accession numbers for complete annotated genes are provided in parentheses.

**Table 1 ijms-22-05580-t001:** Markers derived from annotated *R*-genes within the 21.5 Mbp QTL interval from the reference sequence and mapped to chromosome 1BS in the EGA Blanco/Millewa DH population. Physical position of gene annotations and the genetic map position of *R*-gene markers are shown.

Marker	IWGSC ID	Polymorphism ^a^	Physical Map Position (bp) ^b^	Genetic Map Position (cM)
*1BR_NRC_5A*	TraesCS1B02G002400.1	PAV (EGA Blanco)	1,782,029	7.7
*1B_NRC_A1*	TraesCS1B02G002800.1	PAV (EGA Blanco)	1,846,624	14.4
*1BR_RPP13_1B*	TraesCS1B02G002900.1	PAV (EGA Blanco)	1,943,148	14.4
*1BR_RPP13_2C*	TraesCS1B02G003000.1	PAV (EGA Blanco)	1,971,915	14.4
*1BR_RGA_4B*	TraesCS1B02G003100.1	PAV (EGA Blanco)	2,004,576	14.4
*1BR_RPP13_11D*	TraesCS1B02G003200.1	PAV (EGA Blanco)	2,057,816	10.4
*1BR_RPP13_13E*	TraesCS1B02G003300.1	PAV (EGA Blanco)	2,078,761	10.9
*1BR_RPP13_7F*	TraesCS1B02G003400.1	PAV (EGA Blanco)	2,105,792	10.4
*1B_RGA_E1*	TraesCS1B02G004100.1	PAV (EGA Blanco)	2,357,713	14.9
*1B_NRC_C1*	TraesCS1B02G004600.1	PAV (EGA Blanco)	2,756,985	14.9
*1BR_RR_3A*	TraesCS1B02G004700.1	PAV (EGA Blanco)	2,919,258	14.4
*1BR_LRR_3C*	TraesCS1B02G004900.1	PAV (EGA Blanco)	3,145,388	14.0
*1BR_RR_2C*	TraesCS1B02G005600.1	PAV (Millewa)	3,656,683	7.2
*1B_Lr21_B3*	TraesCS1B02G006300.1	PAV (Millewa)	3,925,448	7.2
*1BR_LRR_1D*	TraesCS1B02G006400.1	PAV (Millewa)	3,935,573	4.0 *
*1BR_LRR_4E*	TraesCS1B02G006800.1	PAV (Millewa)	3,966,850	7.7
*1B_RPM1_A1*	TraesCS1B02G006900.1	PAV (Millewa)	4,049,415	6.8
*1BR_LRR_2F*	TraesCS1B02G007200.1	PAV (Millewa)	4,534,576	7.7
*1BR_LRR_4G*	TraesCS1B02G007300.1	PAV (Millewa)	4,146,127	0 *
*IWB64368*	TraesCS1B02G007800.1	SNP	4,320,651	9.8
*1BR_LRR_1H*	TraesCS1B02G008500.1	PAV (Millewa)	4,483,578	7.7
*1BR_LRR_1I*	TraesCS1B02G008800.1	PAV (Millewa)	4,534,869	7.7
*1BR_LRR_4J*	TraesCS1B02G008900.1	PAV (Millewa)	4,541,238	7.7
*1B_Pm3_F1*	TraesCS1B02G051200.1	PAV (Millewa)	6,272,172	24.8
*1B_RPP13_C1*	TraesCS1B02G014500.1	PAV (EGA Blanco)	7,040,210	28.1
*1B_Pm3_D1*	TraesCS1B02G014800.1	PAV (EGA Blanco)	7,257,001	25.2
*1B_Pm3_D6*	TraesCS1B02G014900.1	PAV (Millewa)	7,253,228	26.6
*1B_Pm3_I1*	TraesCS1B02G042500.1	PAV (Millewa)	22,155,020	42.5

^a^ Marker polymorphism represented by presence/absence variation (PAV) or single nucleotide polymorphism (SNP). Absence of marker (null-allele) in parental variety is shown in brackets; ^b^ physical position of tile from where marker was designed; * marker with segregation distortion. Critical chi-square value >10% (1 degree freedom).

**Table 2 ijms-22-05580-t002:** One-dimension scanning positions of *R*-gene markers associated with QTL for SNB resistance in a DH population derived from the cross of resistant and susceptible parents, EGA Blanco and Millewa, respectively. Gene markers with significant associations (*p* < 0.001) are highlighted in grey shading.

QTL	Marker (Gene)	Polymorphism ^a^	Physical Position (bp) ^b^	Genetic Position (cM) ^c^	LOD ^d^	PVE ^e^	Additive ^f^
*QSnl07.daw-1B*	*1BR_RR_2C*	PAV (Millewa)	3,656,683	7.2	0.1	0.1	0.4
	*1BR_NRC_5A*	PAV (EGA Blanco)	1,782,029	7.7	13.6	20.3	−5.9
	*IWB64368*	SNP	4,320,856	9.8	18.6	25.8	−6.7
	*1BR_RPP13_11D*	PAV (EGA Blanco)	2,057,816	10.4	0.0	0.0	0.0
	*1BR_RPP13_13E*	PAV (EGA Blanco)	2,078,761	10.9	0.0	0.0	0.0
*QSnl08.daw-1B*	*1BR_LRR_3C*	PAV (EGA Blanco)	3,145,388	14.0	0.0	0.0	0.1
	*1BR_RGA_4B*	PAV (EGA Blanco)	2,004,576	14.4	10.0	16.9	−3.9
	X3476283 * (*Snn1*)	PAV (EGA Blanco)	2,357,713	14.9	10.1	17.1	−3.9
	*IWB7273*	SNP	1,203,706	16.3	0.0	0.0	−0.1
	*IWB11788*	SNP	1,203,968	16.8	0.0	0.0	−0.1
*QSnl07.daw-5B*	*IWB12583*	SNP	539,072,791	229.8	0.1	0.2	0.5
	*IWB22904*	SNP	541,343,146	230.7	0.1	0.1	0.4
	*Xfcp623* (*Tsn1*)	PAV	ND	234.9	7.5	10.1	−4.2
	*Xfcp620* (*WAK35*)	PAV	546,568,033	234.9	7.5	10.1	−4.2
	*IWA4103*	SNP	558,625,387	250.8	0.6	0.7	−1.1
	*5B_RGA_SNP_3A1*	SNP	562,793,027	253.6	0.4	0.4	−0.9
*QSnl08.daw-5B*	*Xfcp623* (*Tsn1*)	PAV	ND	234.9	0.3	0.5	−0.6
	*Xfcp620* (*WAK35*)	PAV	546,568,033	234.9	0.3	0.5	−0.6
	*IWA4103*	SNP	558,625,387	250.8	2.8	4.6	−2.0
	*5B_RGA_SNP_3A1*	SNP	562,793,027	253.6	4.3	6.9	−2.4
	*5B_RPP13_5B*	PAV (Millewa)	563,121,363	254.5	0.0	0.0	0.0
	*5B_Dir_1C*	PAV (Millewa)	570,174,000	255.4	0.0	0.0	0.0

^a^ Marker polymorphism represented by Presence/Absence Variation (PAV) or Single Nucleotide Polymorphism (SNP). Absence of gene (null-allele) in parental variety is shown in brackets; ^b^ International Wheat Genome Sequencing Consortium (IWGSC) RefSeq v1.0, bp: base pairs; ^c^ genetic map position derived from the EGA Blanco/Millewa doubled haploid mapping population [33] in centimorgan (cM) distances; ^d^ logarithm of the odds; ^e^ percentage variation of the phenotype explained in the doubled haploid mapping population; ^f^ positive and negative effects indicate the allele was inherited from the male (Millewa) and the female (EGA Blanco) parent, respectively; * diagnostic marker for *Snn1.*

**Table 3 ijms-22-05580-t003:** Markers derived from annotated *R*-genes within the 45.6 Mbp QTL interval from the reference sequence and mapped to chromosome 5BL in the EGA Blanco/Millewa DH population. Physical position of gene annotations and the genetic map position of *R*-gene markers are shown.

Marker	IWGSC ID	Polymorphism ^a^	Physical Map Position (bp) ^b^	Genetic Map Position (cM)
*Xfcp623*	ND ^c^	PAV (EGA Blanco)	ND ^c^	234.9
*Xfcp620*	TraesCS5B02G368200.1	SNP	546,568,033	234.9
*IWA4103*	TraesCS5B02G380700.1	SNP	558,625,387	250.8
*5B_RGA_SNP_3A1*	TraesCS5B02G383800.1	SNP	562,793,027	253.6
*5B_RPP13_5B*	TraesCS5B02G384300.1	PAV (Millewa)	563,121,363	254.5
*5B_WRKY_2A*	TraesCS5B02G384600.1	PAV (EGA Blanco)	563,332,765	255.9
*5B_Dir_1C*	TraesCS5B02G391000.1	PAV (Millewa)	570,174,000	255.4
*5B_LRR_4D_SNP*	TraesCS5B02G393400.1	SNP	571,609,010	258.3
*5B_PiK2_SNP_11_A1*	TraesCS5B02G394100.1	SNP	571,998,218	260.1
*5B_Pi_21_1A*	TraesCS5B02G396100.1	PAV (EGA Blanco)	573,141,867	263
*5B_RPK_2A*	TraesCS5B02G402100.1	PAV (Millewa)	578,607,490	268
*5B_SerThr_4C_SNP*	TraesCS5B02G403300.1	SNP	580,085,277	266.2
*5B_PerOX_2B*	TraesCS5B02G404300.1	PAV (EGA Blanco)	580,431,388	264.4
*5B_PerOX_SNP_1.1*	TraesCS5B02G404200.1	SNP	580,431,742	269.3
*5B_Chi_1A_SNP_2*	TraesCS5B02G403700.1	SNP	589,164,993	270.2

^a^ Polymorphism represented by presence/absence variation (PAV) or single nucleotide polymorphism (SNP) between wheat cultivars EGA Blanco and Millewa. Absence of marker (null-allele) in parental variety is shown in brackets; ^b^ physical position of tile from where marker was designed; ^c^ ND: sequence not identified in the targeted QTL interval of the physical map in the IWGSC reference genome sequence of Chinese Spring.

## Data Availability

The data presented in this study are available on request from the corresponding author.

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
