# Peer review of "Genes Associated with Foliar Resistance to Septoria Nodorum Blotch of Hexaploid Wheat (Triticum aestivum L.)"

_ijms, 2021, doi:10.3390/ijms22115580_

Round 1
Reviewer 1 Report
The work and molecular analysis result look very nice and covered all the molecular work, but this work aims to find a resistance gene for Septoria nodorum blotch (SNB) in wheat. The author needs to mention the pathologist part and cover it with details.
What is the Septoria nodorum race or isolate that has been used?
Did the artificial or natural infection has been used for screening?
What is the susceptible checks line severity look like?
The author needs to include a table of disease severity and infection type results in the manuscript and what scale was used for scoring the disease.
Author Response
Point 1: The work and molecular analysis result look very nice and covered all the molecular work, but this work aims to find a resistance gene for Septoria nodorum blotch (SNB) in wheat. The author needs to mention the pathologist part and cover it with details.
Response 1: It was mentioned in the materials and methods in the original submission that the EGA Blanco/Millewa DH population was accessed from the study described in Francki et al. 2011 (reference #6). However, the reviewer raising issues regarding phenotypic analysis made it obvious that the original manuscript was not clear on what data was used for identifying genes associated with SNB response.
The trial designs, inoculum, inoculation methodology, phenotypic data and statistical data for QTL analysis to detect genes associated in QTL in this study was based on the original phenotypic data described in Francki et al. 2011. In that study, the SNB response and associated statistical analysis was detailed and all issues the reviewer raised below are detailed in that reference. We had used the mean plot SNB scores for individuals of the population in the earlier QTL analysis for genes associated with disease response in this study.
We, therefore, feel that it is unnecessary to report the phenotyping data in this study when detailed analysis for evaluation in 2007 and 2008 was reported in Francki et al. 2011. Instead, we made revisions to the 4.1. Plant material and evaluation for SNB response in the Materials and Methods to explain that details of the phenotyping experiments can be accessed from Francki et al 2011 and the mean plots values for individuals of the DH population for each year was used in QTL analysis for genes associated with SNB response in this study.
We appreciate the reviewer bringing this to our attention so that we were able to clarify this doubtful point.
Point 2: What is the Septoria nodorum race or isolate that has been used?
Response 2: See response 1
Point 3: Did the artificial or natural infection has been used for screening?
Response 3: See response 1
Point 4: What is the susceptible checks line severity look like?
Response 4: See response 1
Point 5: The author needs to include a table of disease severity and infection type results in the manuscript and what scale was used for scoring the disease.
Response 5: See response 1
ALL CHANGES ACCORDING TO THE REVIEWERS’ COMMENTS CAN BE SEEN IN THE ATTACHED REVISED MANUSCRIPT.

Reviewer 2 Report
This study is aimed to align and compare the genetic map representing QTL interval for Septoria nodorum blotch (SNB) foliar resistance on chromosome 1BS and 5BS with the physical map and identify resistance genes associated with SNB response from high-quality reference wheat genome sequence. The research is able to provide some new elements. However, there are few aspects that have to be revised. The study can be considered for publication in IJMS after revisions.
Comments: some examples
Citation: Lastname, F.; Lastname, F.; Lastname, F. Title. Int. J. Mol. Sci. 2021, 22, x. -???
Abstract is a long introduction. Only the last sentence is a result with a conclusion. Please give a 1-2 sentence introduction. One sentence objectives is also needed. Then results with one sentence conclusion.
L57: Begining of the sentence – please use full name of the patogen.
L100: Please give the objective in separate paragraph. Objective is a 4-row complicate sentence.
L107-111: This section is M and M.
L190: Francki et al. (2018) – numbering
Figures 3 and 4 : numbers and letters can not be seen – need to improve the quality of the figures.
L481, L553: different presentation of internet sources.
L670: Nicotiana benthamiana – in italic
L620, L623: Stagnospora nodorum blotch or Septoria nodorum Blotch
Author Response
Response to Reviewer 2 Comments
Point 1: Citation: Lastname, F.; Lastname, F.; Lastname, F. Title. Int. J. Mol. Sci. 2021, 22, x. -???
Response 1: We have included author names in citation in the revised document. The assigned volume and article number and DOI needs to be completed by the editorial office if and when the publication has been accepted.
Point 2: Abstract is a long introduction. Only the last sentence is a result with a conclusion. Please give a 1-2 sentence introduction. One sentence objectives is also needed. Then results with one sentence conclusion.
Response 2: We have revised the abstract according to the reviewers’ suggestion. A two sentence introduction to the work was included followed by a clear and concise statement of the aim of the study. The sentences following the aim summarise the main results whereas the final sentence provides the conclusion for the study.
Point 3: L57: Begining of the sentence – please use full name of the patogen.
Response 3: Changed to Parastagonospora nodorum
Point 4: L100: Please give the objective in separate paragraph. Objective is a 4-row complicate sentence.
Response 4: The objectives of the study was clarified as a separate paragraph as recommended by the reviewer. Moreover, we have shortened the objective statement to be succinct and less complicated.
Point 5: L107-111: This section is M and M.
Response 5: This section has been deleted from L107-111 as suggested by the reviewer and incorporated into the Materials and Methods. The
Point 6: L190: Francki et al. (2018) – numbering
Response 6: References were converted to numbering as per journal requirement. Manuscript checked for similar differences and changed accordingly.
Point 7: Figures 3 and 4 : numbers and letters can not be seen – need to improve the quality of the figures.
Response 7: The fonts and size for numbers and letters have increased and made in bold to improve readability and quality of in Figures 3 and 4. See attached revised document for improved quality.
Point 8: L481, L553: different presentation of internet sources.
Response 8: Internet sources checked and correct as per original submission. The internet source http://www.gramene.org at L481 refers to the Gramene database whereas https://www.geneious.com at L553 refers to the software program for DNA sequence alignments Geneious. Therefore, no change to the original manuscript.
Point 9: L670: Nicotiana benthamiana – in italic
Response 9: Nicotiana benthamiana italicised in the revised document.
Point 10: L620, L623: Stagnospora nodorum blotch or Septoria nodorum Blotch
Response 10: Changed to Septoria nodorum blotch for consistency.
ALL CHANGES ACCORDING TO THE REVIEWERS’ COMMENTS CAN BE SEEN IN THE ATTACHED REVISED MANUSCRIPT.

Round 2
Reviewer 1 Report
The manuscript "Genes associated with foliar resistance to Septoria nodorum blotch of hexaploid wheat (Triticum aestivum L.)" is a very interesting topic targeting a new resistance gene against Septoria nodorum blotch. The Auther cove every side for this study.
I recommend accepting the manuscript as its.